# Daphnetin Alleviates Bleomycin-Induced Pulmonary Fibrosis through Inhibition of Epithelial-to-Mesenchymal Transition and IL-17A

**DOI:** 10.3390/cells12242795

**Published:** 2023-12-08

**Authors:** Soo-Jin Park, Hyung Won Ryu, Ji-Hyeong Kim, Hwa-Jeong Hahn, Hyun-Jae Jang, Sung-Kyun Ko, Sei-Ryang Oh, Hyun-Jun Lee

**Affiliations:** 1Natural Product Research Center, Korea Research Institute of Bioscience and Biotechnology (KRIBB), Cheongju-si 28116, Republic of Korea; soojin921@kribb.re.kr (S.-J.P.); ryuhw@kribb.re.kr (H.W.R.); kimgh0222@kribb.re.kr (J.-H.K.); hwajounghan@kribb.re.kr (H.-J.H.); water815@kribb.re.kr (H.-J.J.); 2Department of Biomolecular Science, University of Science & Technology (UST), Daejeon 34113, Republic of Korea; ksk1230@kribb.re.kr; 3Chemical Biology Research Center, Korea Research Institute of Bioscience and Biotechnology (KRIBB), Cheongju-si 28116, Republic of Korea

**Keywords:** daphnetin, idiopathic pulmonary fibrosis, TGF-β, Th17, IL-17A

## Abstract

Idiopathic pulmonary fibrosis (IPF) is a chronic and refractory interstitial lung disease. Although there is no cure for IPF, the development of drugs with improved efficacy in the treatment of IPF is required. Daphnetin, a natural coumarin derivative, has immunosuppressive, anti-inflammatory, and antioxidant activities. However, its antifibrotic effects have not yet been elucidated. In this study, we investigated the antifibrotic effects of daphnetin on pulmonary fibrosis and the associated molecular mechanism. We examined the effects of daphnetin on splenocytes cultured in Th17 conditions, lung epithelial cells, and a mouse model of bleomycin (BLM)-induced pulmonary fibrosis. We identified that daphnetin inhibited IL-17A production in developing Th17 cells. We also found that daphnetin suppressed epithelial-to-mesenchymal transition (EMT) in TGF-β-treated BEAS2B cells through the regulation of AKT phosphorylation. In BLM-treated mice, the oral administration of daphnetin attenuated lung histopathology and improved lung mechanical functions. Our findings clearly demonstrated that daphnetin inhibited IL-17A and EMT both in vitro and in vivo, thereby protecting against BLM-induced pulmonary fibrosis. Taken together, these results suggest that daphnetin has potent therapeutic effects on lung fibrosis by modulating both Th17 differentiation and the TGF-β signaling pathway, and we thus expect daphnetin to be a drug candidate for the treatment of IPF.

## 1. Introduction

Idiopathic pulmonary fibrosis (IPF) is a serious chronic lung disease accounting for 20% of all interstitial lung diseases (ILDs) in clinical practice [1,2]. As there is no effective cure for IPF, the median survival time of IPF patients is only 2–4 years after diagnosis [3]. In addition to similar symptoms of other ILDs, precise diagnosis becomes more important to confirm different patterns of fibrosis such as high-resolution computed tomography patterns (HRCTs). Based on an accurate diagnosis, further studies on IPF are needed to find appropriate treatment strategies targeting innate and adaptive immune cells, alveolar epithelial cells, and increased proteins in fibrotic lungs [4,5,6]. Despite continuous efforts to develop therapies for IPF, to date, only two drugs have been approved for the treatment of IPF: pirfenidone (PFD) and nintedanib. These two drugs inhibit IPF progression and increase the survival period of IPF but cannot completely cure the disease. Currently, lung transplantation is the only solution for increasing the life expectancy of IPF patients [7]. Therefore, research on new treatments for IPF is essential.

Daphnetin (7,8-dihydroxycoumarin) is a coumarin derivative extracted from the *Daphne* genus, including *Daphne kiusiana*, which is widely distributed in Korea and Japan [8,9]. Research has shown that daphnetin acts as a biologically active substance with anti-inflammation, antiproliferation, antihypoxia, and anticancer properties [8,10]. Additionally, daphnetin has the potential to relieve the symptoms and lung conditions of respiratory diseases through anti-inflammatory effects [11,12]. Daphnetin inhibited LPS-induced acute lung injury and activation of macrophage and human alveolar epithelial cells by suppressing the NF-κB-dependent signaling pathway [11]. Daphnetin treatment also reduced proinflammatory cytokines and inflammatory cell infiltration by inhibiting the JAK2–STAT3 signaling pathway and attenuated severe acute-pancreatitis-associated lung injury [13]. Interestingly, daphnetin contributes to immune balance by suppressing Th17 differentiation and boosting regulatory T (Treg) cell development in colitis [14]. Recently, we reported that the ethyl acetate fraction extracted from *D. kiusiana* has anti-inflammatory effects in lung epithelial cells and alleviates airway inflammation in a cigarette-smoke-exposed COPD mouse model [9]. However, the effect of daphnetin on pulmonary fibrosis has not been investigated.

There is evidence that IL-17 is related to pulmonary fibrosis pathogenesis and progression. IL-17 induces the accumulation of epithelial-to-mesenchymal transition (EMT) and collagen in epithelial cells via increasing TGF-β [11]. In addition, blocking IL-17A signaling may attenuate the profibrotic effects through blocking the TGF-β1 signaling pathways [15]. We previously reported that kurarinone and theophylline produce antifibrotic effects on pulmonary fibrosis by targeting Th17 differentiation and TGF-β signaling [15,16]. Based on these previous studies, we hypothesized that daphnetin may have beneficial effects in IPF treatment, or at least attenuate symptoms, through the suppression of IL-17A production.

Therefore, given the complexity of pulmonary fibrosis, targeting multiple signaling pathways may have advantages for the treatment of the disease. In this study, we evaluated daphnetin as a potential candidate for the treatment of IPF using an in vitro Th17 differentiation system as well as EMT in epithelial cells. We further investigated an in vivo bleomycin (BLM)-induced pulmonary fibrosis mouse model and the underlying mechanism.

## 2. Materials and Methods

### 2.1. Daphnetin

Daphnetin was extracted from *D. kiusiana* as previously described [9,17]. Daphnetin was purified using preparative HPLC and attained purities of over 95% as determined using an ultra-performance liquid chromatography-photodiode array (UPLC-PDA).

### 2.2. Mice

Male Balb/c and DO11.10 TCR transgenic mice were purchased from DBL (Eumseing, South Korea) and Taconic (Rensselaer, NY, USA), respectively. Mice were kept in animal facilities under standard conditions with a dark/light cycle of 12:12 h with adequate water and food. The animals were allowed to adapt to the laboratory environment for at least 7 days prior to the beginning of the experiment. Animal experiment plans were approved by IACUC of KRIBB (Approval number: KRIBB-AEC-23061).

### 2.3. Splenocyte Culture and Compounds Treatment

Primary splenocytes were isolated from DO11.10 TCR transgenic mice and cultured in Th17 differentiation conditions: RPMI-1640 media containing 10% FBS, 50 μM β-mercaptoethanol (M3148, Sigma-Aldrich-Merck, Burlington, MA, USA), 2000 nM OVA_323~339_ peptide (lot#22-45561, Peptron, Daejeon, South Korea), 0.5 ng/mL mouse TGF-β (#763102), and 10 ng/mL murine IL-6 (#575704) (BioLegend, San Diego, CA, USA). Daphnetin, pirfenidone (HY-B0673, Med Chem Express, Monmouth Junction, NJ, USA), and vehicle were treated at the indicated concentrations for 3 days. 

### 2.4. Cell Culture

Human BEAS-2B lung epithelial cells were purchased from the ATCC (Manassas, VA, USA). Cells were cultured at 37 °C in a 5% CO_2_-humidified incubator and cultured in DMEM containing 10% FBS. Cells were treated with daphnetin, pirfenidone, vehicle, and 2 ng/mL human TGF-β (#781802, BioLegend, San Diego, CA, USA) in DMEM containing 0.1% FBS.

### 2.5. Cell Viability Assay

Cells were seeded in 96-well plates and treated with the indicated concentrations of daphnetin. After 24 h or 72 h, cells were incubated with WST (water-soluble tetrazolium salt) for 2 h according to the Cell Counting Kit-8 assay protocol (EZ-3000, DoGenBio, Seoul, South Korea). Then, the optical densities were measured at 450 nm.

### 2.6. BLM-Induced Pulmonary Fibrosis Mouse Model

Eight-week-old male Balb/c mice were assigned into five groups (n = 6 in each group): control + solvent group (NC), bleomycin + solvent group (BLM), BLM + daphnetin (10 mg/kg) group (DAP10), BLM + daphnetin (20 mg/kg) group (DAP20), and BLM + pirfenidone (150 mg/kg) group (PFD). According to a previous study [15], the BLM-induced fibrosis model was induced via an intratracheal instillation of BLM (4 mg/kg body weight, B1141000, Sigma-Aldrich Inc). Daphnetin and pirfenidone were dissolved in a solvent composed of 10% (*v*/*v*) DMAC (D137510), 10% (*v*/*v*) TWEEN80 (P8074, Sigma-Aldrich Inc.) and 80% (*v*/*v*) HPBCD (H0979, Tokyo Chemical Industry, Tokyo, Japan). Daphnetin and pirfenidone were administered orally six times a week for 2 weeks. The control and bleomycin groups received solvent only. After lung function testing, mice were sacrificed using the standard cervical dislocation method, and then lung tissue was collected for further analysis. Animal experiments were replicated twice. A schematic representation of the experimental design is shown in Figure 3A.

### 2.7. Pulmonary Mechanical Function Test

On day 28, mice were anesthetized with an intraperitoneal administration of pentobarbital sodium (100 mg/kg, ENTOBAR^®^, Hanlim Pharm. Co., Ltd., Seoul, South Korea). Mice were then tracheostomized to insert an 18 G cannula into trachea and connected to the small animal ventilator (flexiVent, SCIREQ, Inc., Montreal, QC, Canada). The following parameters of lung mechanics were evaluated: resistance (Rrs), elastance (Ers), and compliance (Crs) using Snapshot-150; lung tissue damping (G) and lung tissue elastance (H) using Quick Prime-3; static compliance (Cst) using the pressure–volume curve. The average of these three measurements was used for the analysis.

### 2.8. Staining for Histopathological Analysis

Left lung tissues were fixed in 10% formalin, embedded in paraffin, and sectioned into a thickness of 4 μm. For hematoxylin and eosin staining, sections were deparaffinized and rehydrated. Sections were stained with hematoxylin and eosin solution (ab245880, Abcam, Waltham, MA, USA) and dehydrated. For Masson’s trichrome staining, sections were stained with Bouin’s fluid, Weigert’s iron hematoxylin, Biebrich scarlet/acid fuchsin solution, phosphomolybdic/phosphotungstic acid solution, aniline blue solution, and acetic acid solution (ab150686, Abcam, Waltham, MA, USA). Sections were dehydrated and mounted. The samples were observed using a microscope.

### 2.9. Hydroxyproline Assay

To quantify collagen, hydroxyproline detection in lung lysates was measured using a Hydroxyproline Colorimetric assay kit (K555-100, BioVision, Milpitas, CA, USA) as described previously [15].

### 2.10. Quantitative PCR Analysis

Total RNA was extracted, and the cDNA was prepared as described previously [18]. For real-time PCR, SYBR Green qPCR PreMIX (RT500M, Enzynomics, Daejeon, Korea) and QuantStudio 1 Real-Time PCR (Applied Biosystems, Waltham, MA, USA) were used. The relative gene expression levels were evaluated using their ratio to GAPDH mRNA or β-actin mRNA.

### 2.11. Western Blot Analysis

Lysates were prepared and Western blot was performed as described previously [15]. Primary antibodies, recognizing pAkt (#2965), pSmad2/3 (#8828), pERK (#9101), p-p38 (#9211), p38 (#9212), ColIα1(#84336), E-cadherin (#3195), N-cadherin (#4061) (primary antibody dilution was 1:1000, Cell Signaling Technology, Danvers, MA, USA), Akt (#610861), Smad2/3 (#610843), RORγt (#562197) (1:1000, BD Biosciences, San Jose, CA, USA), ERK (sc-514302, 1:1000, Santa Cruz Biotechnology, Dallas, TX, USA), αSMA (1:2000, Sigma-Aldrich Inc.), and β-actin (#664802, 1:5000, BioLegend), were used. Protein band densities were analyzed using ImageJ version 1.52a (National Institutes of Health, Bethesda, MD, USA). Target protein and internal control protein bands were selected using the rectangle tool, and the area inside the mountains were calculated using a magic wand. Relative target protein levels were calculated by dividing target protein value into internal control protein value.

### 2.12. Measurement of Cytokines by ELISA

The amounts of IL-17A in cell supernatants and lung homogenates were measured using an ELISA Kit (88-7371-88, Invitrogen, Thermo Fisher Scientific, Waltham, MA, USA). The amount of total TGF-β1 in lung homogenates was analyzed using ELISA kits (DY1679-05, R&D System Inc., Minneapolis, MN, USA). Lung homogenates were prepared in PBS containing a protease inhibitor cocktail and 1% Triton X-100. We coated 96-well plates (NUNC) overnight at 4 °C with capture antibodies. On the next day, plates were blocked for 1 h with blocking buffer. Cell supernatants and lung homogenates were added to the wells and incubated for 2 h. Detection antibody was then incubated for 1 h and avidin-HRP was added for 30 min. Plates were then incubated with substrate solution and stop solution was added. Absorbance was read at 450 nM.

### 2.13. Statistical Analysis

All data analysis was performed using GraphPad Prism software (ver. 6.07, GraphPad Software, Inc., San Diego, CA, USA). Data are presented as means ± standard error of the mean (SEM). A one-way analysis of variance (ANOVA) was used depending on the number of independent variables. Additionally, one-way ANOVA was performed using Turkey’s multiple comparison test for multiple comparisons. All presented data are indicative of a minimum of three independent determinations and passed normality tests. *p* < 0.05 was considered statistically significant.

## 3. Results

### 3.1. Daphnetin Inhibits Th17 Differentiation by Suppressing Transcription Factor RORγt

Several studies have reported a relation between IL-17 and IPF. Thus, we assessed the inhibitory effect of daphnetin on Th17 differentiation [19,20]. When evaluated under Th17 differentiation conditions, daphnetin effectively reduced IL-17A secretion in a dose-dependent manner with no cytotoxicity (Figure 1A,B). The inhibitory effect of daphnetin (*p* < 0.001 compared with the untreated control) was more potent than the control drug PFD at 10 μM (*p* < 0.01 compared with the untreated control). Thus, we focused on daphnetin as a major functional compound inhibiting IL-17A production. 

The transcription factor RORγt is a master regulator of Th17 differentiation and is induced in activated T cells under Th17 differentiation conditions [21]. To understand the mechanism of daphnetin on IL-17A inhibition, we investigated the expression levels of RORγt and IL-17A in differentiated Th17 cells. Treatment of daphnetin significantly reduced the mRNA levels of IL-17A (Figure 1C). Furthermore, the mRNA and protein levels of RORγt were markedly inhibited by daphnetin treatment (Figure 1C,D). These results indicate that daphnetin significantly inhibited IL-17A production by repressing the master transcription factor, RORγt, which is essential for Th17 differentiation.

### 3.2. Daphnetin Reduces TGF-β-Induced EMT by Inhibiting AKT Signaling Pathway in BEAS-2B Cells

EMT is a process of the loss of epithelial proteins leading to a shift towards a more mesenchymal phenotype and is one of the characteristics of fibrosis [22]. We investigated the effect of daphnetin on TGF-β-induced EMT in BEAS-2B cells. Daphnetin treatment significantly reduced the expression of the EMT markers, Col1α1 and αSMA, induced by TGF-β in a concentration-dependent manner (Figure 2A). Next, we examined the effect of daphnetin on the TGF-β signaling pathway. The levels of p-Smad2/3, p-AKT, p-p38, and p-ERK were elevated by TGF-β. Daphnetin significantly reduced the levels of p-AKT and the levels of p-Smad2/3, but to a lesser extent. On the contrary, the levels of p-p38 were increased by daphnetin in TGF-β-treated BEAS-2B cells (Figure 2B). Daphnetin did not affect the viability of the cells in these conditions (Figure 2C). Daphnetin notably inhibited TGF-β-induced Col1α1 and αSMA expression. Collectively, these data suggest that daphnetin inhibits TGF-β-induced EMT primarily by regulating the AKT signaling pathway and therefore may have the potential to inhibit pulmonary fibrosis.

### 3.3. Daphnetin Improves Mechanical Function of Lung in BLM-Induced Pulmonary Fibrosis

We next explored the effect of daphnetin on lung fibrosis using a BLM-induced fibrosis mouse model. To assess the therapeutic effect of daphnetin, we administered daphnetin from day 14 to day 27 after BLM treatment (Figure 3A). BLM treatment significantly increased relative lung weight. Oral administration of 10 and 20 mg/kg of daphnetin reduced the relative lung weight in BLM-treated mice (Figure 3B). Next, the mechanics of the respiratory system were measured to identify the effect of daphnetin on BLM-induced lung function using a flexiVent system. BLM treatment increased the value of resistance (Rrs), elastance (Ers and H), and damping of tissue (G). On the other hand, BLM instillation reduced lung volume, distensibility (Cst), and compliance (Crs). Oral administration of daphnetin restored BLM-induced mechanical dysfunction to an extent comparable to PFD (Figure 3C,D). Daphnetin treatment markedly inhibited BLM-increased lung weight and resistance. These data demonstrate that daphnetin restored the impaired lung mechanics in BLM-induced pulmonary fibrosis.

### 3.4. Daphnetin Attenuates BLM-Induced Fibrotic Changes in Lung Tissue

To evaluate the effect of daphnetin on the fibrotic changes of lung tissues, histological alterations were assessed using H&E and Masson’s trichrome stain. BLM administration increased histological changes such as inflammatory cell infiltration, destruction of lung structure, and collagen deposition. Oral administration of daphnetin markedly attenuated BLM-induced fibrotic changes in lung tissue (Figure 4A,B). Moreover, daphnetin treatment quantitatively suppressed BLM-increased collagen production (Figure 4C). Next, to investigate the antifibrotic effect of daphnetin in BLM-treated mice, the expression levels of profibrotic markers in lung tissue were analyzed. Daphnetin administration attenuated the expression of profibrotic markers induced by BLM (Figure 4D,E). BLM increased N-cadherin and decreased E-cadherin, and daphnetin reversed these changes induced by BLM as effectively as PFD (Figure 4E). Phosphorylated Smad2/3, AKT, and ERK were reduced in the lung lysates of the daphnetin-treated group compared to the BLM-treated group (Figure 4F). Daphnetin treatment also suppressed BLM-induced TGF-β production (Figure 4G). In addition, IL-6 was secreted from damaged epithelial cells in the early stages of fibrosis and is required for Th17 differentiation [23,24]. Daphnetin treatment significantly decreased the expression levels of IL-6 in lung tissue induced by BLM (Figure 4H). Collectively, daphnetin treatment effectively inhibited BLM-induced collagen production in mice, suggesting that daphnetin attenuates BLM-induced fibrotic changes in lung tissue by inhibiting the TGF-β signaling pathway in BLM-induced lung fibrosis.

### 3.5. Daphnetin Suppresses IL-17A Production and Th17 Differentiation in BLM-Induced Pulmonary Fibrosis

Our results of in vitro experiments showed that daphnetin inhibited IL-17A production and Th17 differentiation. We therefore investigated whether daphnetin has the same effect in vivo. In BLM-induced pulmonary fibrosis, daphnetin significantly inhibited the production of IL-17A protein and mRNA induced by BLM in lung tissue (Figure 5A,B). In the BLM group, the mRNA and protein levels of RORγt were increased, while in daphnetin treatment groups, the levels of RORγt showed a reduction (Figure 5C,D). The IL-17A inhibitory effect of daphnetin was similar to that of PFD. Together, these results indicate that the antifibrotic effects of daphnetin on BLM-induced lung fibrosis may be attributed to the inhibition of IL-17A production and Th17 differentiation.

## 4. Discussion

IPF is the most common type of interstitial respiratory disease without an identifiable cause. At present, the exact mechanisms of damage in IPF are still poorly understood. PFD and nintedanib are commonly used for the treatment of IPF, but these drugs cannot completely cure the disease or improve the overall survival rate of patients [16]. The purpose of IPF management is primarily to improve the patient’s respiratory function and to stabilize or reduce disease progression. Extracts form herbs, trees, and flowers have been widely used as conventional therapies in Asia [25]. In recent years, many traditional medicine ingredients have been used in alternative medicine to treat various diseases because of their extensive effects and low toxicity [26]. Hence, there is a strong need to develop new products from natural resources with fewer side effects than existing therapies for the effective treatment of IPF. In this study, we demonstrated that daphnetin, derived from traditional medicinal herbs, has therapeutic effects on lung fibrosis, and we also defined the underlying mechanism.

Daphnetin has various pharmacological activities, such as anti-inflammation and antioxidation, and has been clinically used for treating lumbago, rheumatoid arthritis, and coagulation disorders [26]. Several studies have reported that daphnetin has a significant protective effect against LPS-induced lung inflammation and injury via the regulation of NF-κB-driven proinflammatory signaling and the JAK/STATs pathway [11,27]. Furthermore, daphnetin induced Nrf2-dependent antioxidant responses and reduced arsenic-induced cytotoxicity and apoptosis in human lung epithelial cells [12]. However, the antifibrotic effects of daphnetin on BLM-induced fibrosis are still not clear.

IL-17A plays a key role in tissue inflammation by promoting the production of proinflammatory factors [28]. In addition, IL-17A regulates various downstream pathways such as the TGF-β, Akt, Smad, and MAPKs signaling pathways. Furthermore, adverse IL-17A signaling could cause tissue damage and cell death, resulting in tissue fibrosis [29]. In recent years, various studies have increasingly recognized the significance of IL-17A as a therapeutic target for IPF. IL-17A was detected in the bronchoalveolar lavage fluid and tissues of patients with IPF [30]. In the early stages of the fibrosis process, BLM increases the production of IL-17A in CD4+ T cells and γδT cells, which contributes to the development of lung fibrosis [31,32]. IL-17A induces EMT and myofibroblast differentiation through upregulation of Acta2, Col1a1, Snail family, and fibronectin [30]. Furthermore, blocking IL-17A using IL-17A antibody treatment alleviates acute inflammation and fibrotic changes in mice [16]. Therefore, IL-17A plays important roles throughout the early inflammatory response and the late fibrotic process in IPF progression. Reducing IL-17A is one of the important therapeutic concepts. We found that *D. kiusiana* has the inhibitory effect on IL-17A production among numerous natural products in differentiated Th17 cells. With further analysis, we identified daphnetin as a biologically functional compound with an inhibitory effect on IL-17A secretion. It is important to note that the inhibitory effect of daphnetin was stronger than that of other compounds from which it was derived. We also showed that daphnetin reduced the transcription factors RORγt, the master regulator of Th17 differentiation. These results suggest that the inhibitory effects of daphnetin on IL-17A production could contribute to the antifibrotic effects, especially through the regulation of Th17 differentiation. Further studies, including flow cytometry and in-depth molecular analysis, are needed to confirm the role of daphnetin on Th17 differentiation.

TGF-β is a potent profibrotic cytokine and master switch in fibrotic responses in various organs [15]. TGF-β promotes EMT, fibroblast activation, and ECM accumulation, leading to the development of fibrosis in IPF patients [33]. Consequently, targeting TGF-β-induced EMT may be an effective method to inhibit the progression of pulmonary fibrosis. Using our experimental system, daphnetin suppressed TGF-β-induced profibrotic changes in lung epithelial cells. TGF-β regulates EMT through Smad as well as non-Smad signaling pathways, such as AKT, p38, and ERK [15]. Our results showed that daphnetin mainly reduced the phosphorylation of AKT, which could affect the TGF-β-dependent activation of inflammatory cells as well as the epithelial cells in BLM-treated mice. Furthermore, IL-6 and TGF-β are required to induce the production of IL-17A and RORγt, in addition to the activation of epithelial cells and fibroblasts [23,24]. The levels of TGF-β and IL-6 were significantly inhibited by daphnetin treatment compared to those in BLM-treated mice, and, as a result, daphnetin regulated the activation of epithelial cells as well as Th17 differentiation. In the fibrotic response, fibroblasts and epithelial cells produce ECM proteins including collagen, elastin, and fibronectin, and serve pivotal roles in tissue fibrosis [34]. Hydroxyproline is a major component of collagen produced by both fibroblasts and epithelial cells, and quantification of hydroxyproline plays an important role in measuring the progression of fibrosis [35]. Daphnetin treatment significantly inhibited BLM-induced hydroxyproline as well as ColIα1 expression in lung tissues, indicating that daphnetin might reduce the collagen production in fibroblasts and epithelial cells in the progression of fibrotic responses.

BLM causes acute inflammation, including cytokine production and excessive recruitment of inflammatory cells into lung tissue, which last up to 7 days. Fibrotic changes, with fibroblasts’ activation and extracellular matrix deposition, reach a peak around day 14 [36]. There have been numerous reports about the preventive effects of natural products, typically administered within first 7 days after BLM treatment. However, it is important to distinguish between anti-inflammatory and antifibrotic effects. The therapeutic effects during fibrotic phase are much more favorable for fibrosis treatment and prolonging the lifespan of IPF patients. Therefore, to estimate the antifibrotic effects on the fibrotic phase, we administrated daphnetin from day 14 following BLM exposure. As shown in our results, daphnetin effectively restored lung function and fibrotic changes in BLM-treated mice, suggesting the therapeutic potential of daphnetin for pulmonary fibrosis.

However, this study has limitations. Despite the therapeutic effect of daphnetin on pulmonary fibrosis, isolating and purifying a large amount of daphnetin from natural sources is inefficient. Therefore, in order to develop daphnetin as a therapeutic agent, it is important to establish a process to synthesize daphnetin.

Nevertheless, in this study, we identified, for the first time, the therapeutic effects of daphnetin on lung fibrosis by targeting multiple pathways. Given the complex nature of IPF pathogenesis, the need for compounds against multiple targets is constantly increasing in the development of new drugs for fibrosis [37].

In conclusion, we demonstrated that daphnetin has antifibrotic effects on pulmonary fibrosis in both in vitro and in vivo studies. Our results showed that daphnetin treatment ameliorated BLM-induced fibrotic changes and respiratory dysfunction. Furthermore, daphnetin exerts its therapeutic effects by suppressing TGF-β downstream signaling pathways and Th17 differentiation. Therefore, our findings provide new insights into the development of daphnetin for chronic lung disease as well as IPF.

## Figures and Tables

**Figure 1 cells-12-02795-f001:**
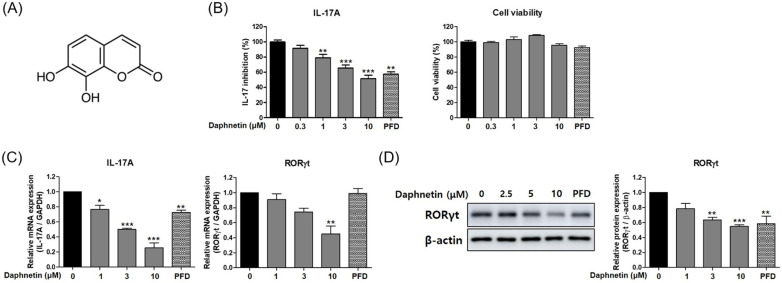
Daphnetin inhibits IL-17A production via Th17 differentiation pathway in vitro. (**A**) Chemical structure of daphnetin. (**B**) The levels of IL-17A and cytotoxicity in differentiated Th17 cells with daphnetin or vehicle. (**C**) Relative mRNA levels of IL-17A and RORγt in differentiated Th17 cells treated with daphnetin or vehicle. (**D**) Protein levels of RORγt in differentiated Th17 cells treated with daphnetin or vehicle. The relative band density of target protein in a graph was normalized to β-actin and calculated using ImageJ. All data are presented as the mean ± SEM of three independent experiments; * *p* < 0.05, ** *p* < 0.01, *** *p* < 0.001 compared with the untreated control.

**Figure 2 cells-12-02795-f002:**
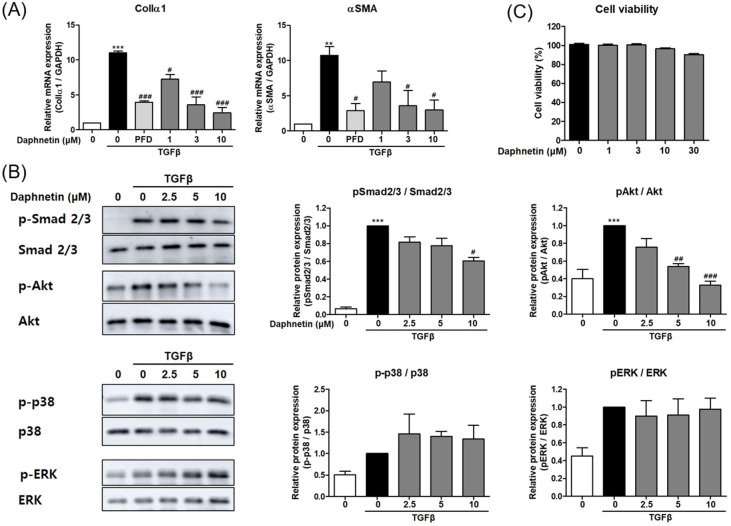
Daphnetin suppresses TGF-β signaling pathways in BEAS-2B cells. (**A**) Relative mRNA expression of ColIα1 and αSMA in BEAS-2B cells treated with TGF-β (2 ng/mL, 24 h) with daphnetin or vehicle. (**B**) The levels of phosphorylated Smad2/3, AKT, ERK, and p38 in BEAS-2B cells treated with TGF-β (2 ng/mL for 1 h) with daphnetin or vehicle. The relative band density of target protein in a graph was normalized to β-actin and calculated using ImageJ. (**C**) Cytotoxicity of daphnetin in BEAS-2B cells. All data are presented as the mean ± SEM of three independent experiments; ** *p* < 0.01, *** *p* < 0.001, compared with untreated control; ^#^ *p* < 0.05, ^##^ *p* < 0.01, ^###^ *p* < 0.001, compared with TGF-β-treated group.

**Figure 3 cells-12-02795-f003:**
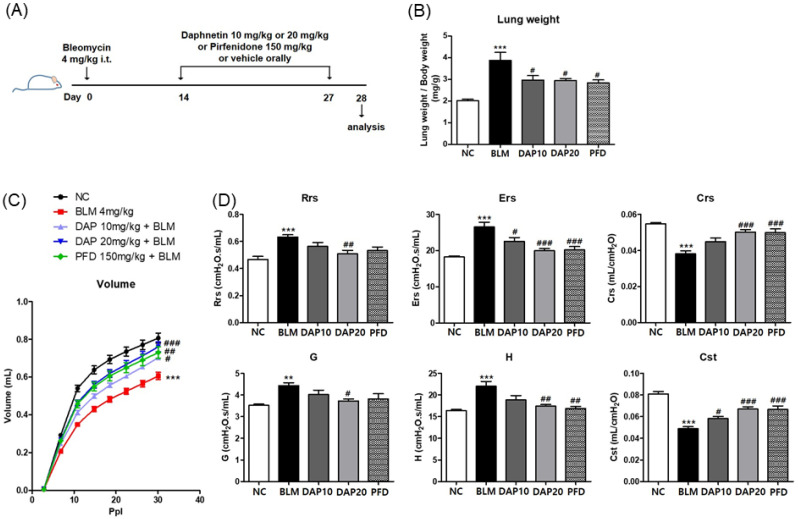
Daphnetin improves BLM-induced pulmonary function. (**A**) Scheme of the experimental design. (**B**) Relative lung weight (left lung weight to body weight) of each mouse. (**C**,**D**) Pulmonary function parameters, the values of pressure–volume curves, Rrs, Ers, Crs, G, H, and Cst, were measured using a flexiVent system. NC, control + solvent group; BLM, BLM + solvent group; DAP10 and DAP20, BLM + daphnetin (10 and 20 mg/kg) groups; PFD, BLM + pirfenidone (150 mg/kg) group. All data are presented as the mean ± SEM from 6 mice for each group; ** *p* < 0.01, *** *p* < 0.001, compared with NC; ^#^
*p* < 0.05, ^##^
*p* < 0.01, ^###^
*p* < 0.001, compared with BLM.

**Figure 4 cells-12-02795-f004:**
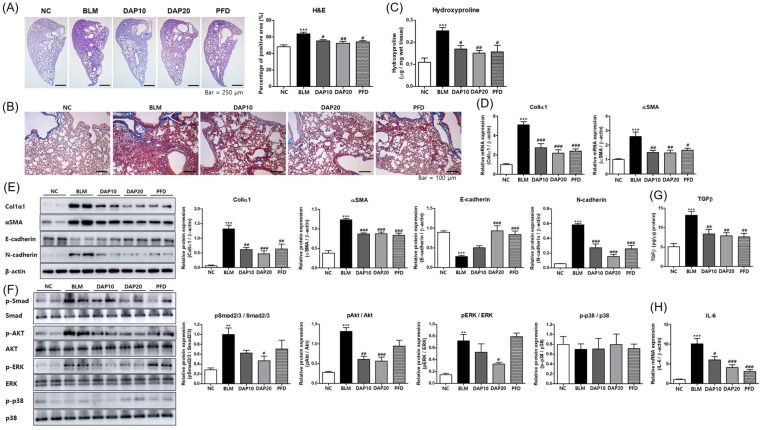
Daphnetin attenuates BLM-induced fibrotic changes in lung tissue. (**A**) H&E staining of lung sections from experimental groups. Densities of an image were quantitated using ImageJ. (**B**) Masson’s trichrome staining of lung sections from each group with collagen staining blue. (**C**) The content of hydroxyproline in lung of mice. (**D**) The mRNA expression of ColIα1 and αSMA in lung tissues. (**E**) Western blot analysis of ColIα1, E-cadherin, and N-cadherin in lung tissues. (**F**) Western blot analysis of phosphorylated Smad2/3, AKT, ERK, and p38 in lung tissue. The relative band density of target protein in a graph was normalized to β-actin and calculated using ImageJ. (**G**) The levels of TGF-β1 in lung of each group. (**G**) Western blot analysis of phosphorylated Smad2/3, AKT, ERK, and p38 in lung tissue. (**H**) The mRNA expression of IL-6 in lung tissues. All data are presented as the mean ± SEM from 6 mice for each group; ** *p* < 0.01, *** *p* < 0.001, compared with NC; ^#^
*p* < 0.05, ^##^
*p* < 0.01, ^###^
*p* < 0.001, compared with BLM.

**Figure 5 cells-12-02795-f005:**
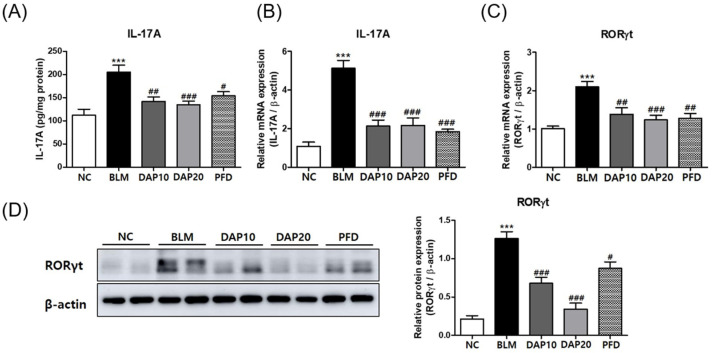
Daphnetin suppresses IL-17A production and Th17 differentiation in lung tissue. (**A**) The levels of IL-17A in lung homogenates. (**B**,**C**) Expression levels of IL-17A and RORγt in lung tissues. (**D**) Western blot analysis of RORγt in lung tissue from experimental groups. The relative band density of target protein in a graph was normalized to β-actin and calculated using ImageJ. All data are presented as the mean ± SEM from 6 mice for each group; *** *p* < 0.001, compared with NC; ^#^
*p* < 0.05, ^##^
*p* < 0.01, ^###^
*p* < 0.001, compared with BLM.

## Data Availability

Data are contained within the article or Appendix A.

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
