# Peer review of "Daphnetin Alleviates Bleomycin-Induced Pulmonary Fibrosis through Inhibition of Epithelial-to-Mesenchymal Transition and IL-17A"

_cells, 2023, doi:10.3390/cells12242795_

Round 1

Reviewer 1 Report

Comments and Suggestions for Authors

The study by Soo-Jin Park et al. focuses on addressing idiopathic pulmonary fibrosis (IPF), a chronic and challenging interstitial lung disease for which a cure has not yet been found. The research focus is based on dafnetin, a natural coumarin-derived compound that has been shown to possess immunosuppressive, anti-inflammatory and antioxidant properties. The study investigates in depth the antifibrotic effects of dafnetin and its associated molecular mechanism. To this end, a series of experiments were carried out involving the evaluation of dafnetin in splenocytes cultured under Th17 conditions, lung epithelial cells and a mouse model of bleomycin-induced pulmonary fibrosis (BLM). The results obtained are interesting and revealed that dafnetin was able to inhibit the production of the cytokine IL-17 in developing Th17 cells. Furthermore, dafnetin was shown to suppress epithelial-mesenchymal transition (EMT) in TGF-β-treated BEAS2B cells. In a setting of BLM-treated mice, oral administration of dafnetin reduced lung histopathology and improved lung mechanical functions. Taken together, these findings suggest that dafnetin possesses promising therapeutic effects in the context of pulmonary fibrosis. Modulation of Th17 differentiation and the TGF-β signaling pathway emerge as key underlying mechanisms. This study raises dafnetin as a potential candidate for the treatment of IPF, offering an exciting prospect for future research in this area. However, I have some minor caveats regarding the submitted manuscript.

1.       In the introduction section, the information on dafnetin is relevant and well presented. However, it would be helpful to provide a brief description of previous studies related to dafnetin and its effect on lung diseases or related conditions. This would help to contextualize the focus of the current research.

2.       In general, authors should provide the catalog number of all reagents and kits used in the study for reproducibility purposes.

3.       In section 2.5 "Cell viability assay", a brief description of how the assay was performed could be added, in addition to mentioning the manufacturer's protocol, and highlighting the number of cells seeded in each well used for this analysis.

4.       Some procedures could benefit from more clarity. For example, in section 2.3. Splenocyte Culture and Compounds Treatment, a brief description of how the assay was performed, what culture conditions were used for Th17 differentiation, what methodology was used to validate Th17 differentiation, reference number 11 provided in this section does not correspond because the methodology for Splenocyte isolation was not provided in this referenced manuscript.

5.       In section 2.8. Staining for Histopathological Analysis, a brief description of how the assay was performed could be added, in addition to mentioning the manufacturer's protocol.

6.       In section 2.11 "Western Blot Analysis", you mention that the densities of the protein bands were analyzed with ImageJ. The authors could provide a brief description of how this analysis was performed and whether internal controls were used.

7.       In section 2.12, the authors mention the measurement of IL-17A and TGF-β1 by ELISA kits. Reference number 11 is used as a reference for the methodology used. However, the referenced manuscript does not adequately describe the methodology used. It would be beneficial to provide details on the sample preparation and the specific procedure of the ELISA assays according to the manufacturer's protocol.

8.       The authors mention in the materials and methods section to have performed the calculation of the band density of the western blot results, however, the results obtained from this relative quantification are not shown. I suggest that the authors perform a quantification of the band density of the western blot results presented in all the figures, in order to calculate the statistical difference between each of the study groups, observing this relative quantification in a graph would enrich the study and improve the understanding of the results for the readers.

Reviewer 2 Report

Comments and Suggestions for Authors

The topic is HOT. The manuscript is quite well written. I have some suggestions:

1) Abstract. Idiopathic pulmonary fibrosis (IPF) is a chronic and refractory interstitial lung disease. Although there is no cure for IPF, the development of drugs with improved efficacy in treatment of IPF is required. Daphnetin, a natural coumarin derivative, has immunosuppressive, anti-inflammatory, and anti-oxidant activities. However, its anti-fibrotic effects have not yet been elucidated. In this study, we investigated the anti-fibrotic effects of daphnetin on pulmonary fibrosis and associated molecular mechanism. We examined the effects of daphnetin on splenocytes cultured in Th17 conditions, lung epithelial cells, and a mouse model of bleomycin (BLM)-induced pulmonary fibrosis. Daphnetin was administrated orally to mice after intratracheal injection of BLM, and lung function, histological analysis, and expression of profibrotic markers were evaluated. We identified that daphnetin inhibited IL-17 production in developing Th17 cells. We also showed that daphnetin suppressed epithelial-to-mesenchymal transition (EMT) in TGF-β-treated BEAS2B cells through regulation of AKT phosphorylation. In BLM-treated mice, oral administration of daphnetin attenuated lung histopathology and improved lung mechanical functions. Taken together, these results suggest that daphnetin has potent therapeutic effects on lung fibrosis by modulating both Th17 differentiation and the TGF-β signaling pathway, and thus we expect that daphnetin may be a potential drug candidate for treatment of IPF.  It might be beneficial to include a sentence in the abstract that briefly summarizes the key findings of the study. This can provide readers with a quick overview of the research. 

2) 1. Introduction 33 Idiopathic pulmonary fibrosis (IPF) is a serious chronic lung disease [1]. As there is 34 no effective cure for IPF, the median survival time of IPF patients is only 2-4 years after 35 diagnosis [2]. Despite continuous efforts to develop therapies for IPF, to date there are 36 only two drugs have been approved for the treatment of IPF: pirfenidone (PFD) and 37 nintedanib. These two drugs inhibit IPF progression and increase the survival period of 38 IPF, but cannot completely cure the disease. Currently, lung transplantation is the only 39 solution to increasing the life expectancy of IPF patients [3]. Therefore, research on new 40 treatments for IPF is essential. 

I suggest that you include some information in order to complete the manuscript. Below you can find some works that could give useful ideas in expanding this part, I suggest:

a- Regeneration or Repair? The Role of Alveolar Epithelial Cells in the Pathogenesis of Idiopathic Pulmonary Fibrosis (IPF). Cells. 2022 Jun 30;11(13):2095. doi: 10.3390/cells11132095.

b- Macrophage Implication in IPF: Updates on Immune, Epigenetic, and Metabolic Pathways. Cells 202312, 2193. https://doi.org/10.3390/cells12172193.

c- Caveolin-1-Related Intervention for Fibrotic Lung Diseases. Cells 202312, 554. https://doi.org/10.3390/cells12040554.

d- Evaluation of Correlations between Genetic Variants and High-Resolution Computed Tomography Patterns in Idiopathic Pulmonary Fibrosis. Diagnostics (Basel). 2021 Apr 23;11(5):762. doi: 10.3390/diagnostics11050762. 

3) Therefore, in this study, we identified daphnetin as a potential candidate for treat-61 ment with IPF using in vitro Th17 differentiation system and the bleomycin (BLM)-in-62 duced pulmonary fibrosis mouse model. Please, re-write the paragraph and undeline the novelty of the study and the aim.

4) 2.13. Statistical Analysis. Improve the description of the statistical test used.

5)  3. Results. Please, underline the most important statistically significant data to support and clarify the results

6) 4. Discussion 246 247 IPF is the most common type of interstitial respiratory diseases without an identifia- 248 ble cause. At present, the exact mechanisms of damage in IPF are still poorly understood. 249 PFD and nintedanib are commonly used for the treatment of IPF, but these drugs cannot 250 completely cure the disease or improve the overall survival rate of patients [11]. Please, summarise here the most important results of the study to clarify the discussion.

7) In conclusion, we demonstrated that daphnetin has anti-fibrotic effects on pulmonary 309 fibrosis both in vitro and in vivo studies. Our results showed that daphnetin treatment 310 ameliorated BLM-induced fibrotic changes and respiratory dysfunction. Furthermore, 311 daphnetin exerts its therapeutic effects by suppressing TGF-β downstream signaling path-312 ways and Th17 differentiation. Therefore, our findings may provide new insights into the 313 development of daphnetin for chronic lung disease as well as IPF. Underline the novelty of the paper and the possible clinical implications.

Comments on the Quality of English Language

Minor changes of English language are required

Reviewer 3 Report

Comments and Suggestions for Authors

Soo-Jin Park et al in this manuscript, describe the therapeutic effect of daphnetin in pulmonary fibrosis through PCR and Wb analysis of cell lines, as well as corroboration in the bleomycin (BLM)-induced pulmonary fibrosis mouse model where Daphnetin is shown to decrease collagen damage and deposition through inhibition of EMT and decrease of IL-17.

The manuscript is scientifically sound and the methodology is necessary to conclude what they say, however, there are some minor points corrections.

Introduction is complete

methodology

1 for reproducibility of the readers it is recommended that the authors describe the methodology a little bit more.

2 add the concentration of antibodies.

Results

1 Authors are encouraged to add the reference of the cell lines 142.

2 It is recommended to justify the use of the cell line, for readers not specialized in the disease and interested in daphnetin.

3 Homogenize IL17A or IL17.

Discussion

1 In lines 280-288, it is not clear if they talk about the present manuscript or another one where they tested Kiusiana (add reference).

2 Add a reference to lines 301 and 302.

3 Add limitations of the study.

Reviewer 4 Report

Comments and Suggestions for Authors

The manuscript by Park and colleagues is an interesting study investigating the anti-fibrotic potential of daphnetin, a natural derivative of coumarin. To this aim, the Authors administered daphnetin to a mouse model of bleomycin-induced pulmonary fibrosis, showing that the global lung detrimental effect of BLM was attenuated. Furthermore, they showed in BEAS2B cells that daphnetin coud suppress EMT, a recognized mechanism at the basis of fibrosis. This well-written article deals with a relevant subject, and the observations suggested by the Authors are certainly worthy of note.  

The following is a synopsis of concerns raised, based on the data presented on the manuscript.

1. Abstract: very clear, nothing to report.

2. Materials and methods: I understand that the Authors refer to procedures described in other papers. However, for the sake of clarity and to ensure full reproducibility, I suggest them to describe the methods followed in greater detail. 

Line 92: please add a space between "2" and "weeks"

Line 92: how did the Authors sacrifice the mice?

Line 98: what about the pentobarbital sodium concentration used?

Line 116 (qPCR analysis): I suggest the Authors to add the primer sequences. 

Line 132: ELISA kit was specific for active of total TFG-beta1? Please specify.

3. Results:

Line 145: please add the statistical comparison regarding the p value reported in brackets in the figure.

Line 149: please add at least one reference supporting the role of RORgammaT.

Line 166: please add at least one reference supporting what stated here (EMF as one of the characteristics of fibrosis).

Fig. 2: did the Authors investigate the levels of aSMA as a protein (via WB), in addition to mRNA transcript?

Fig. 4: please specify how the percentage of positive area was determined, thank you.

Fig. 4-C: is it possible that the hydroxyproline levels decreased because of the impact of daphnetin in fibroblast differentiation? Please discuss this point. 

Fig. 4-C: did the Authors investigate the amount of IL-6, which is known to be related to Th17 differentiation?

Line 240: I'd be more prudent in asserting that Th17 differentiation was impacted, since no flow cytometry analysis supporting this observation was carried out. Or, additional intracellular/extracellular markers concerning Th17 differentiation should be investigated, including IL-17F, RORalpha, CCR4, CCR6. 

Round 2

Reviewer 2 Report

Comments and Suggestions for Authors

The manuscript has been improved as requested, I have no further comments

Comments on the Quality of English Language

Minor changes of English language are required